# The effect of hydration status on plasma FGF21 concentrations in humans: A subanalysis of a randomised crossover trial

**Harriet A. Carroll** [1,2]ʘ *, **Yung-Chih Chen** [3,4]ʘ, **Iain Templeman** [1], **Lewis J. James** [5], **James A. Betts** [1], **William V. Trim** [1]

**1** Department for Health, University of Bath, Bath, United Kingdom, **2** Cardiovascular Research–Hypertension, Clinical Research Centre, Lund University, Malmö, Sweden, **3** Department of Physical Education, National Taiwan Normal University, Taipei, Taiwan, **4** Institute for Research Excellence in Learning Science, National Taiwan Normal University, Taipei, Taiwan, **5** School of Sport, Exercise and Health Sciences, Loughborough University, Loughborough, United Kingdom

ʘ These authors contributed equally to this work.
\* hc12591@my.bristol.ac.uk

**Data Availability Statement:** All relevant data are within the paper and its Supporting Information files.

## Abstract

### Aim

Fibroblast growth factor 21 (FGF21) has recently been implicated in thirst in rodent models. The mechanisms for this are currently uncertain, and it is unclear whether hydration status can alter FGF21 concentrations, potentially providing an additional mechanism by which hypohydration induces thirst. The aim of this study is therefore to understand whether hydration status can alter circulating FGF21 in humans.

### Methods

Using a heat tent and fluid restriction, we induced hypohydration (1.9% body mass loss) in 16 healthy participants (n = 8 men), and compared their glycaemic regulation to a rehydration protocol (heat tent and fluid replacement) in a randomised crossover design.

### Results

After the hypohydration procedure, urine specific gravity, urine and serum osmolality, and plasma copeptin (as a marker for arginine vasopressin) increased as expected, with no change after the rehydration protocol. In the fasted state, the median paired difference in plasma FGF21 concentrations from the rehydrated to hypohydrated trial arm was -37 (interquartile range -125, 10) pg·mL$^{-1}$ ($P$ = 0.278), with average concentrations being 458 ± 462 pg·mL$^{-1}$ after hypohydration and 467 ± 438 pg·mL$^{-1}$ after rehydration; mean difference -9 ± 173 pg·mL$^{-1}$.

### Conclusion

To our knowledge, these are the first causal data in humans investigating hydration and FGF21, demonstrating that an acute bout of hypohydration does not impact fasted plasma

**Funding:** HAC is funded by the Economic and Social Research Council (grant no.: ES/J50015X/1). HAC has received conference fees, travel and accommodation (2016, 2017, 2018, and 2019) from Danone Nutricia, and has received funding from Esther Olssons II & Anna Jönssons Minnesfond foundations. JAB has received funding from the BBSRC, GlaxoSmithKline, Lucozade Ribena Suntory, Kellogg's, Nestlé, and PepsiCo. The funders had no role in study design, data collection and analysis, decision to publish, or preparation of the paper.

**Competing interests:** HAC has received conference fees, travel and accommodation (2016, 2017, 2018, and 2019) from Danone Nutricia, has run a Tate and Lyle sponsored study, and has received funding from Esther Olssons II & Anna Jönssons Minnesfond foundations. JAB has received funding from the BBSRC, GlaxoSmithKline, Lucozade Ribena Suntory, Kellogg's, Nestlé, and PepsiCo. JAB is a scientific advisor to the International Life Sciences Institute (ILSI). This does not alter our adherence to PLOS ONE policies on sharing data and materials. There are no patents, products in development or marketed products associated with this research to declare.

FGF21 concentrations. These data may suggest that whilst previous research has found FGF21 administration can induce thirst and drinking behaviours, a physiological state implicated in increased thirst (hypohydration) does not appear to impact plasma FGF21 concentrations in humans.

## Introduction

Fibroblast growth factor 21 (FGF21) is a peptide hormone synthesised and secreted by several organs (mainly the liver) and is implicated in growth and energy balance [1]. Recent research has provided preliminary evidence that FGF21 might have a role in regulating thirst and drinking behaviours in rodents [2,3]. These behaviours are controlled neurologically: hypertonicity is detected by the central nervous system, relaying this information to the lamina terminalis which contains stretch-sensitive ion channels in areas such as the organum vasculosum [4]. This osmoreception stimulates neurons in the paraventricular nucleus and hypothalamus and results in the secretion of arginine vasopressin (AVP) [5,6], facilitating reductions in renal water losses thus mitigating further losses in body water. During hypohydration, AVP is secreted into circulation from the hypothalamus in order to reduce urinary water losses, thus aiding in blood pressure regulation by maintaining blood volume [5]. Simultaneously, the elevated extracellular osmolality stimulates thirst to encourage fluid ingestion or water-seeking behaviour [7]. Arginine vasopressin is part of the hypothalamic-pituitary-adrenal (HPA) axis. Under stress, AVP potentiates the effects of corticoptropin-releasing hormone (CRH) on adrenocorticotropic hormone and cortisol secretion [8,9] from the pituitary and adrenal glands, respectively.

Fibroblast growth factor 21 has been shown to stimulate the HPA axis in response to prolonged fasting in mice [10,11]. However, FGF21 may also act to reduce AVP in the suprachiasmatic nucleus, [11] meaning the role of FGF21 in thirst could be via pathways which are independent of AVP or hydration status. Rodent studies have demonstrated increased fluid ingestion after FGF21 administration [2,3,12] or a preference for plain water over sweetened water in FGF21-transgenic mice [13,14].

In rodents with free access to chow and water, intravenous infusion of PF-05231023 (a long-lasting FGF21 analogue) did not alter plasma osmolality, but did increase water intake by ~20% after 48 h, with no change in food intake [2]. Urinary output also increased after PF-05231023 infusion including greater absolute excretion of electrolytes. Interestingly this increase in urinary output was greatest (~40% higher than control rats) after 24 h, suggesting FGF21 caused increased urination, which resulted in greater water intake (perhaps indicative of higher thirst). Conversely, PF-05231023 infusion did not have these effects in fluid restricted (hypohydrated) rats. Despite the theoretical relationship between FGF21 and the HPA axis, no differences were seen between treatment and control rats in hypothalamic AVP or CRH expression, concordant with other similar research [3].

Considering FGF21 alters fluid balance, it is perhaps unsurprising that it has also been implicated in blood pressure in accordance with other hormones related to fluid balance (e.g. AVP). In PF-05231023-infused rats, both heart rate and blood pressure increased compared to control rats [2], though this has not been replicated in mice [3]. Such findings indicate that FGF21 may have a role in raising blood pressure in a euhydrated state, causing increased urination which increases thirst and ultimately results in fluid ingestion; this chain of events is currently unclear since polydipsia in mice does not appear to be secondary to polyuria, thus

warranting further investigation [3]. It is unlikely that this cascade is mediated by the kidneys as β-klotho (an FGF21 cofactor involved in FGF21 activity and receptor binding) is expressed only in very low concentrations in the kidneys [2]. Thus, FGF21-mediated drinking responses appear to bypass typical thirst regulatory pathways, such as increased plasma osmolality and AVP secretion.

However, there is evidence that the effect of FGF21 in rodents may be different to primates. For example, in rodents, food intake does not change with PF-05231023 infusion [2], whereas in cynomolgus monkeys and humans food intake decreases after treatment [15]. There is, however, a noted lack of causal human data [16], though recent work has shown PF-05231023 infusion in humans with obesity and hypertriglyceridaemia to increase blood pressure [17], corroborating findings in rodents. In humans, mild hypohydration ($< 2\%$ body mass fluctuation) is commonly achieved during daily living [18]; thus it is of interest to understand whether hydration status can impact FGF21. Whilst Song *et al.* [3] found FGF21 to be unaffected after water deprivation in mice, the relationship between FGF21 and hydration status in humans is currently unknown. As FGF21 may be involved in adipocyte glucose uptake [19], and in aiding in insulin sensitivity and glycaemic regulation [1], there may be important therapeutic benefits to understanding whether FGF21 can be manipulated by hydration status.

We recently conducted a randomised crossover trial investigating the effect of hydration status on glycaemic regulation, finding mild (~2%) hypohydration did not impact glycaemia or insulinaemia *versus* euhydration in healthy adults [20]. After completion of the study, we were able to measure FGF21 concentrations in the remaining fasted plasma samples which we aim to describe herein. Due to the findings of previous research showing FGF21 appears to induce dehydration (e.g. increased urination), we hypothesised that hypohydration (HYPO) would lead to a decrease in fasted plasma FGF21 concentrations compared to when participants were rehydrated (RE), due to water losses suppressing secretory diuretic hormones.

## Materials and methods

### Participants

Participants were recruited using posters at the University of Bath (South West England), which is where the research was conducted, between June 2016 and January 2017. The inclusion criteria required participants to be healthy and not taking prescribed medication or supplements, with the exception of hormonal contraceptives. Therefore, our exclusion criteria were: age $< 18$ y or $\geq 60$ y, self-reported metabolic disease (no body mass restrictions, except self-reported weight loss $> 5$ kg in previous 6 mo), drug dependence, and pregnancy/breastfeeding. All participants gave fully informed consent to undergo the protocol and for their samples and data to be used as appropriate for the project.

We recruited sixteen healthy adults who successfully participated in the study after giving fully informed consent (Table 1). Participants were not offered any incentive to participate, beyond their individualised feedback; considering this, the affluence of the city of Bath, and our inclusion criteria focused on healthy adults, our sample is representative of healthy

**Table 1. Participant characteristics (n = 16).**

| Participant characteristic | Mean ± standard deviation |
| --- | --- |
| Sex (n female) | 8 |
| Age (y) | 30 ± 9 |
| Body mass (kg) | 71.7 ± 9.6 |
| Body mass index (kg/m$^2$) | 24.0 ± 3.4 |

middle-class younger adults (both men and women), interested in their health. Ethical approval was granted from the NHS Health Research Authority, reference: 16/SW/0057 (trial registration: Clinicaltrials.gov: NCT02841449; Open Science Framework: osf.io/ptq7m). Participants were randomised via simple randomisation to be in either the hypohydrated (HYPO) trial arm, or rehydrated (RE) trial arm first.

### Study design

The methods have been previously described in full [20]. This was a randomised crossover trial with a 5–35 day washout period. Three days prior to the intervention, participants replicated their diet (including fluids) and activity patterns. On the third day, participants were instructed to consume $\geq$ 40 mL·kg$^{-1}$ lean body mass of non-alcoholic fluid to ensure euhydration prior to starting the protocol [20].

Following these pre-trial controls, participants came to the laboratory after overnight food and fluid abstention and had a fasted (baseline) blood sample drawn in a euhydrated state, before undergoing a dehydration protocol via 60 min in a heat tent (~45˚C) wearing a sweat suit (RDX EVA Nylon Sauna Sweat Suit). Participants then consumed a sandwich containing $\geq$ 1 g salt to maximise fluid retention and serum osmolality changes. For HYPO, 3 mL·kg$^{-1}$ of plain water was provided, compared to 40 mL·kg$^{-1}$ lean body mass plus 150% water losses from the heat tent procedure for RE. Participants were only permitted to consume low water content foods and the water provided (i.e. no other fluids). Diet and activity were replicated on the subsequent trial arm, with the exception of water intake.

The next day, participants arrived at the laboratory after overnight food and fluid abstention. A fasted arterialised-venous blood sample was obtained from an antecubital vein which was used to assess circulating FGF21 concentrations, followed by a two hour oral glucose tolerance test [20] and a series of appetite tasks [21]. Participants then had a 5–35 day washout period before repeating the protocol under the other intervention arm.

### Biochemical analysis

Biochemical analysis for the primary analytes of the study have previously been described [20]. Fibroblast growth factor 21 was measured via electrochemiluminescence (MSD, Mesoscale Diagnostics LLC; in-house coefficient of variation 5.4%).

### Statistical analysis

Data were checked for normality visually via histogram of the raw data, and PP and QQ plots of the standardised residuals, as previously described [20]. The FGF21 data were not normally distributed, therefore the Wilcoxon test was used for inferential purposes. Spearman's *rho* was conducted to understand the association between fasted plasma FGF21 concentrations and change from baseline in body mass, urine osmolality, serum osmolality, plasma copeptin concentration, serum glucose and insulin concentrations, and age, in order to determine if there were any factors which might be associated with a stronger FGF21 response. All analyses had an alpha level of $\leq$ 0.05.

### Results

Participants lost 1.9 ± 1.2% body mass after HYPO with weight stability after RE [20]. This body mass loss was accompanied by expected hypohydration-induced increases in urine specific gravity, urine osmolality, serum osmolality, and plasma copeptin concentration (as a marker of AVP) and a reduction in cross-sectional muscle area (as a proxy for cell volume)

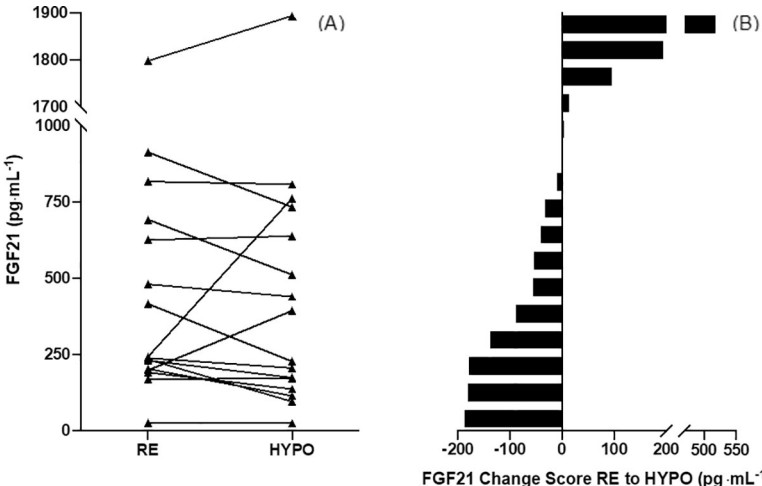

**Fig 1.** (A) Individual fasted FGF21 concentrations RE compared to HYPO. (B) Individual change scores from RE to HYPO in order of magnitude of change. Abbreviations: FGF21, fibroblast growth factor 21; HYPO, hypohydrated trial arm; RE, rehydrated trial arm.

[20]. Compared to RE, HYPO did not alter fasted or postprandial serum glucose or insulin concentrations, or plasma ACTH or cortisol concentrations [20].

The median paired difference from RE to HYPO was -37 pg·mL$^{-1}$ (IQR -125, 10 pg·mL$^{-1}$; $P$ = 0.278; Fig 1). The average plasma FGF21 concentration during HYPO was 458 ± 462 pg·mL$^{-1}$, compared to 467 ± 438 pg·mL$^{-1}$ during RE; mean paired difference -9 ± 173 pg·mL$^{-1}$. There were no associations with fasted plasma FGF21 concentrations and change in any of the hydration or metabolic factors explored (Table 2).

Of the 16 participants, five (aged 22–44 y, median 25 y; n = 4 women) showed an increased in plasma FGF21 concentrations during HYPO compared to RE (ranging from Δ 4 to 518 pg·mL$^{-1}$, median Δ 96 pg·mL$^{-1}$). Comparatively, the remaining 11 participants (aged 22–50 y, median 25 y; n = 4 women) showed a decreased in plasma FGF21 concentrations during HYPO compared to RE (ranging from Δ -188 to -0.2 pg·mL$^{-1}$, median Δ -56 pg·mL$^{-1}$). There were no obvious trends in the data that appeared to explain whether participants responded as per our hypothesis or not.

**Table 2. Associations between fasted plasma FGF21 concentrations in a hypohydrated and euhydrated state and change from baseline in fasted markers of hydration status and metabolic health.**

| | Hypohydrated trial arm | | Euhydrated trial arm | |
|---|---|---|---|---|
| | Correlation coefficient | *P*-value | Correlation coefficient | *P*-value |
| Age | -0.068 | 0.802 | 0.108 | 0.691 |
| Δ Body mass | -0.259 | 0.333 | 0.277 | 0.298 |
| Δ Urine osmolality | -0.216 | 0.421 | -0.097 | 0.721 |
| Δ Serum osmolality | 0.126 | 0.641 | -0.145 | 0.592 |
| Δ Plasma copeptin | -0.107 | 0.704 | 0.221 | 0.428 |
| Δ Serum glucose | 0.306 | 0.249 | -0.127 | 0.640 |
| Δ Serum insulin | -0.494 | 0.052 | -0.481 | 0.059 |

Analyses conducted using Spearman's *rho*. Abbreviations: FGF21, fibroblast growth factor 21.

## Discussion

Contrary to our hypothesis, hypohydration did not decrease fasted plasma FGF21 concentrations. As thirst and desire for higher water content foods was higher during HYPO [21] we found no evidence that a mechanism by which hypohydration is implicated in drinking behaviours in humans is via FGF21 secretion. In rodent models, FGF21 altered drinking behaviours likely via increased urine output. Combined with our findings in humans, this potentially highlights that the relationship is a unidirectional effect. In other words, whilst FGF21 can alter ingestive behaviours resulting in some physiological changes seen during euhydration (i.e. increased urinary output), manipulating hydration itself does not appear to result in differences in plasma FGF21 concentrations.

Alternatively, there could be a causal effect in humans, but this requires a stronger stimulus than the modest level of hypohydration we achieved (~2% body mass loss; serum osmolality 296 mOsm/kg HYPO *versus* 286 mOsm/kg RE), and/or requires chronic fluid deprivation. Such chronic adaptive effects on FGF21 would accord with starvation models in humans whereby FGF21 elevations occur after ~7 days [22]. Whilst the mechanisms surrounding starvation and fluid deprivation with regard to FGF21 responses may differ, this comparison perhaps highlights a role of FGF21 in chronic adaptations—something our study was not designed to measure—thus warranting further exploration in humans.

Additionally, FGF21 theoretically can stimulate the HPA axis, particularly under stress such as prolonged fasting [10]. In our original study, after HYPO, AVP (as measured by copeptin) was elevated to levels seen in those with type 2 diabetes and the metabolic syndrome [23,24], with no difference in FGF21, yet again highlighting the unidirectionality of this relationship compared to the rodent literature. No differences were found in plasma ACTH or cortisol concentrations [20], thus we are unable to make inferences as to the relationship between these hormones and FGF21 in the context of hydration status. We also explored other factors which may moderate circulating FGF21 concentrations, such as age and the level of hypohydration, finding no association, though our conclusions are limited by the relatively small sample size.

In relation to metabolic states, FGF21 may respond specifically to hypermetabolic conditions whereby total energy expenditure has increase resulting in an increase in energy intake. This in turn drives thirst and therefore drinking behaviour [25]. Thus an increase in thirst and drinking behaviours may be a result of elevations in FGF21 (which increases energy expenditure in mice [26]). Our data provide support for this pathway as hydration status itself did not alter thirst or biomarkers of hydration status.

The main aim of the original study was to investigate the effect of hydration status on fasted and postprandial glycaemia, ultimately finding a null effect [20]. As FGF21 is implicated in improved insulin sensitivity and glucose tolerance [1,19], it is perhaps unsurprising in this context that no differences were found. Nonetheless, future work should investigate the postprandial FGF21 response according to hydration status; this is particularly important considering our present analysis was decided *post hoc* and therefore not pre-specified in our research plan or trial registration. This is particularly important as previous work has suggested the thirst-inducing effects of FGF21 to be context-specific, i.e. most prominent in conditions known to induce hypohydration such as prolonged fasting or during a ketogenic diet [3]. It therefore seems paradoxical that dehydrating dietary conditions can amplify the effects of FGF21 but hypohydration itself does not appear to impact FGF21 concentrations.

The initial study was powered to investigate the glycaemic response to hypohydration compared to euhydration. It may be that there is a small but meaningful response, which we may not have been able to detect due to lack of statistical power for FGF21 as the outcome; equally, there may be unmeasured characteristics resulting in responders or non-responders. As such,

the large variance between-participants should be further investigated in larger samples as some concentrations were at levels that are perhaps found in those with obesity or diabetes. Additionally, the kit we used to analyse FGF21 does not specify whether total or active FGF21 is measured; considering its calibration using full-length FGF21 with a predicted molecular mass of 20 kDa, it is most likely we measured active FGF21, which has been shown to the primary circulating form of FGF21 [27].

To our knowledge, this is the first study to examine the causal role of hydration status on circulating FGF21 concentrations in humans. Hydration status did not appear to affect fasted plasma FGF21 concentrations, suggesting that the evidence from rodent FGF21 infusion studies is unidirectional (i.e. FGF21 can cause dehydration, but not *vice versa*). Due to the small sample size, these findings should be taken as preliminary and used as a basis to build new hypotheses pertaining to FGF21-hydration status interactions. Thus, further research should investigate whether FGF21 is altered by hydration status in larger samples, in both the fasted and fed state, as well as investigate whether this is another mechanism (beyond osmolality and AVP changes) implicated in human drinking behaviours. Additionally, future work should aim to answer current outstanding questions such as whether there is a differential FGF21 response to hydration status with a greater degree of hypohydration, and whether this is influenced by the duration of hypohydration.

## Supporting information

**S1 Data.**
(XLSX)

## Acknowledgments

We would like to thank Professor Dylan Thompson for providing the FGF21 kits.

## Author Contributions

**Conceptualization:** Harriet A. Carroll, Yung-Chih Chen, Lewis J. James, James A. Betts, William V. Trim.

**Data curation:** Harriet A. Carroll, Yung-Chih Chen, William V. Trim.

**Formal analysis:** Harriet A. Carroll, William V. Trim.

**Funding acquisition:** Harriet A. Carroll.

**Investigation:** Harriet A. Carroll, Yung-Chih Chen, Iain Templeman.

**Methodology:** Harriet A. Carroll, Lewis J. James, James A. Betts, William V. Trim.

**Project administration:** Harriet A. Carroll, Yung-Chih Chen.

**Resources:** William V. Trim.

**Supervision:** Lewis J. James, James A. Betts.

**Validation:** William V. Trim.

**Visualization:** William V. Trim.

**Writing – original draft:** Harriet A. Carroll, Yung-Chih Chen, William V. Trim.

**Writing – review & editing:** Harriet A. Carroll, Yung-Chih Chen, Iain Templeman, Lewis J. James, James A. Betts, William V. Trim.

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
