## [Decision Letter · Decision Letter 0]

7 Apr 2020

PONE-D-20-06810

The effect of hydration status on plasma FGF21 concentrations in humans

PLOS ONE

Dear Dr Carroll,

Thank you for submitting your manuscript to PLOS ONE. After careful consideration, we feel that it has merit but does not fully meet PLOS ONE’s publication criteria as it currently stands. Therefore, we invite you to submit a revised version of the manuscript that addresses the points raised during the review process.

We would appreciate receiving your revised manuscript by 5th May 2020. To enhance the reproducibility of your results, we recommend that if applicable you deposit your laboratory protocols in protocols.io, where a protocol can be assigned its own identifier (DOI) such that it can be cited independently in the future. For instructions see: http://journals.plos.org/plosone/s/submission-guidelines#loc-laboratory-protocols

We look forward to receiving your revised manuscript.

Kind regards,

Jo Edward Lewis, Ph.D

Academic Editor

PLOS ONE

Journal Requirements:

1. In your Methods section, please provide additional information about the participant recruitment method and the demographic details of your participants. Please ensure you have provided sufficient details to replicate the analyses such as: a) the recruitment date range (month and year), b) a description of any inclusion/exclusion criteria that were applied to participant recruitment, c) a table of relevant demographic details, d) a statement as to whether your sample can be considered representative of a larger population, e) a description of how participants were recruited, and f) descriptions of where participants were recruited and where the research took place.

2. Please provide additional details regarding participant consent. In the ethics statement in the Methods and online submission information, please specify whether trial participants consented to the use of samples in additional analyses (not included by the original trial). If not, please specify whether you obtained separate ethics approval for this analysis.

3. Thank you for including the following funding information within the acknowledgements section; "This work was supported by the Economic and Social Research Council (grant no.: ES/J50015X/1) and the European Hydration Institute Graduate Research Grant. "

"HAC is funded by the Economic and Social Research Council (grant no.: ES/J50015X/1). The funders had no role in study design, data collection and analysis, decision to publish, or preparation of the manuscript."

Additional Editor Comments (if provided):

7/04/2020

Dear Dr Carroll,

PONE-D-20-06810

The effect of hydration status on plasma FGF21 concentrations in humans

Many thanks for sending us your manuscript. It has now been reviewed by two experts in the field and by one of our Editors, and I am pleased to tell you that we found it potentially suitable for publication. You will see however that the referees have raised a number of points that still require your consideration. I would therefore like to offer you the opportunity to revise this manuscript taking these comments into account. You should explain the changes that you have made in response to the referees’ comments in a covering letter. Please also make the changes suggested by our editorial staff to bring your manuscript into line with house style – this will speed subsequent processing of your paper if it is accepted.

Please submit an electronic version of your revised paper within 28 days of receipt of this letter. If new experiments have been requested which will be impossible to complete within this time, we will consider an appeal to extend this deadline. Please contact the Editorial Office if you have any queries.

As I am sure you will appreciate, I am unable as yet to make any final commitment regarding acceptance of this manuscript.

Yours sincerely,

Dr Jo Edward Lewis

Reviewers' comments:

Reviewer's Responses to Questions

**Comments to the Author**

1. Is the manuscript technically sound, and do the data support the conclusions?

Reviewer #1: Yes

Reviewer #2: Yes

2. Has the statistical analysis been performed appropriately and rigorously? 

Reviewer #1: Yes

Reviewer #2: Yes

3. Have the authors made all data underlying the findings in their manuscript fully available?

Reviewer #1: Yes

Reviewer #2: Yes

4. Is the manuscript presented in an intelligible fashion and written in standard English?

Reviewer #1: Yes

Reviewer #2: Yes

5. Review Comments to the Author

Reviewer #1: An important study with broadly interesting results regarding FGF21 involvement in fluid balance in humans.

A few minor points. On line 75/76 authors state that PF-05231023 reduces blood pressure, but the manuscript they cite shows the reverse. On line 78 the Song reference is stated to have used rats, but the species used in this study were mice.

Regarding the hypothesis in the introduction, wouldn't it be more logical to expect that heat induced water loss from sweat would suppress secretion of a diuretic factor from the liver?

Major point:

Would correcting FGF21 levels for plasma volume depletion affect these results? (i.e. to albumin or whatever is standard).

Reviewer #2: This work by Carroll and colleagues explores the effect of hydration on circulating levels of the endocrine factor FGF21 in humans. Previous work in rodents has suggested that FGF21 could regulate thirst and the authors examined whether plasma FGF21 levels were physiologically altered by hydration status. These data are important for the field and warrant publication, but a few minor issues need to be addressed beforehand.

1) It would be worthwhile to include in the introduction and discussion that FGF21 has been shown to increase fluid intake primarily under potentially hypermetabolic conditions. It has been proposed that this increase in fluid intake in response to FGF21 is actually due to increases in food intake (i.e., prandial thirst) (BonDurant and Potthoff; PMID: 29727594).

2) FGF21 does not increase fluid intake under conditions of high sucrose (von Holstein-Rathlou et al.; PMID: 26724858). This citation was also omitted on the sweet taste reference.

3) A potential issue is how FGF21 levels were measured. The authors need to provide more information about this electrochemiluminescence assay. What part of FGF21 does this assay measure. As the authors may know, FGF21 is cleaved and inactivated on the C-terminus. Thus, did the authors measure total or “active” FGF21? These issues should also be addressed in the discussion as potential limitation of the results.

6. PLOS authors have the option to publish the peer review history of their article (what does this mean?). If published, this will include your full peer review and any attached files.

Reviewer #1: Yes: Matthew Gillum

Reviewer #2: No

---

## [Author Response · Author response to Decision Letter 0]

16 May 2020

[Editor comment] Please ensure that your manuscript meets PLOS ONE's style requirements, including those for file naming. The PLOS ONE style templates can be found at

[Author response] We have checked the style requirements and believe we have met the criteria. 

[Editor comment] 1. In your Methods section, please provide additional information about the participant recruitment method and the demographic details of your participants. Please ensure you have provided sufficient details to replicate the analyses such as: a) the recruitment date range (month and year), b) a description of any inclusion/exclusion criteria that were applied to participant recruitment, c) a table of relevant demographic details, d) a statement as to whether your sample can be considered representative of a larger population, e) a description of how participants were recruited, and f) descriptions of where participants were recruited and where the research took place.

[Author response] We have added subsections to our methods, the first of which is “Participants”. Within this section we have included the information requested:

“Participants

Participants were recruited using posters at the University of Bath (South West England), which is where the research was conducted, between June 2016 and January 2017. The inclusion criteria required participants to be healthy and not taking prescribed medication or supplements, with the exception of hormonal contraceptives. Therefore, our exclusion criteria were: age < 18 y or ≥ 60 y, self-reported metabolic disease (no body mass restrictions, except self-reported weight loss > 5 kg in previous 6 mo), drug dependence, and pregnancy/breastfeeding. 

We recruited sixteen healthy adults who successfully participated in the study after giving fully informed consent (Table 1). Participants were not offered any incentive to participate, beyond their individualised feedback; considering this, the affluence of the city of Bath, and our inclusion criteria focused on healthy adults, our sample is representative of healthy middle-class younger adults (both men and women), interested in their health.” (lines 124-129)

Additionally, we have included a table with key participant characteristics: age, sex weight, BMI.

[Editor comment] 2. Please provide additional details regarding participant consent. In the ethics statement in the Methods and online submission information, please specify whether trial participants consented to the use of samples in additional analyses (not included by the original trial). If not, please specify whether you obtained separate ethics approval for this analysis.

[Author response] We obtained fully informed consent to conduct all the necessary procedures on participants, such as cannulation and taking a pre-specified amount of blood. Participants consented for their samples to be used for relevant analyses, but we had not specified exactly what these analyses would be. Therefore, FGF21 analyses were consensually covered and did not require further approval or consent. As FGF21 was unplanned, it was not included on our trial registration unlike the main analytes, such as glucose and insulin. We have added the following to the manuscript:

“All participants gave fully informed consent to undergo the protocol and for their samples and data to be used as appropriate for the project.” (lines 121-122)

[Editor comment] 3. Thank you for including the following funding information within the acknowledgements section; "This work was supported by the Economic and Social Research Council (grant no.: ES/J50015X/1) and the European Hydration Institute Graduate Research Grant. "

"HAC is funded by the Economic and Social Research Council (grant no.: ES/J50015X/1). The funders had no role in study design, data collection and analysis, decision to publish, or preparation of the manuscript."

[Author response] We have removed this from the manuscript and would appreciate it being added to our funding statement. Apologies for the mix-up. 

Reviewers' comments

[Reviewer comment] Reviewer #1: An important study with broadly interesting results regarding FGF21 involvement in fluid balance in humans.

A few minor points. On line 75/76 authors state that PF-05231023 reduces blood pressure, but the manuscript they cite shows the reverse. On line 78 the Song reference is stated to have used rats, but the species used in this study were mice.

[Author response] Thank you for highlighting these errors, we have amended as appropriate (changes in red):

“though recent work has shown PF-05231023 infusion in humans with obesity and hypertriglyceridaemia to reduce blood pressure, corroborating findings in rodents.” (lines 94-95)

“Whilst Song et al.3 found FGF21 to be unaffected after water deprivation in smice…” (line 98)

[Reviewer comment] Regarding the hypothesis in the introduction, wouldn't it be more logical to expect that heat induced water loss from sweat would suppress secretion of a diuretic factor from the liver?

[Author response] Thank you for this insight. Since there was no causal human work to go from, our hypothesis was built on reverse causality from the findings of previous research. However, your hypothesis fits the causal direction of our study design better and makes logical sense. We have therefore amended our hypothesis and discussion as follows (changes in red):

“Due to the findings of previous research showing FGF21 appears to induce dehydration (e.g. increased urination), we hypothesised that hypohydration (HYPO) would lead to a decrease in fasted plasma FGF21 concentrations compared to when participants were rehydrated (RE), due to water losses suppressing secretory diuretic hormones.” (lines 110-112)

“Contrary to our hypothesis, hypohydration did not decrease fasted plasma FGF21 concentrations.” (line 219)

Major point:

[Reviewer comment] Would correcting FGF21 levels for plasma volume depletion affect these results? (i.e. to albumin or whatever is standard).

[Author response] We agree that correcting for plasma volume would be interesting, and as such we did measure plasma volume using haematocrit and haemoglobin. However, we made a methodological error in standardising posture properly during the study, which meant our plasma volume data were not reliable. We outlined this in our original manuscript of the main study findings (doi: 10.1152/japplphysiol.00771.2018) as well as providing the (unreliable) data with an explanatory note in our published data for others to scrutinise (https://researchdata.bath.ac.uk/id/eprint/547). 

Whilst mechanistically interesting to correct for plasma volume, uncorrected data still provide valid results as most studies do not measure plasma volume. Nonetheless, we do appreciate these would have been valuable data to have had. 

[Reviewer comment] Reviewer #2: This work by Carroll and colleagues explores the effect of hydration on circulating levels of the endocrine factor FGF21 in humans. Previous work in rodents has suggested that FGF21 could regulate thirst and the authors examined whether plasma FGF21 levels were physiologically altered by hydration status. These data are important for the field and warrant publication, but a few minor issues need to be addressed beforehand.

[Author response] We appreciate your overall positive review and hope we have addressed the issues you raised below. 

[Reviewer comment] 1) It would be worthwhile to include in the introduction and discussion that FGF21 has been shown to increase fluid intake primarily under potentially hypermetabolic conditions. It has been proposed that this increase in fluid intake in response to FGF21 is actually due to increases in food intake (i.e., prandial thirst) (BonDurant and Potthoff; PMID: 29727594).

[Author response] Thank you for this article and idea. Since understanding the interaction between energy expenditure, food, and thirst was not the aim of the study, we have acknowledged this theory in our discussion only, as follows:

“In relation to metabolic states, FGF21 may respond specifically to hypermetabolic conditions whereby total energy expenditure has increase resulting in an increase in energy intake. This in turn drives thirst and therefore drinking behaviour [25]. Thus an increase in thirst and drinking behaviours may be a result of elevations in FGF21 (which increases energy expenditure in mice [26]). Our data provide support for this pathway as hydration status itself did not alter thirst or biomarkers of hydration status.” (lines 249-254)

[Reviewer comment] 2) FGF21 does not increase fluid intake under conditions of high sucrose (von Holstein-Rathlou et al.; PMID: 26724858). This citation was also omitted on the sweet taste reference.

[Author response] Apologies for not including this; we have added the citation to the introduction regarding sweet taste (line 64). However, we are unsure of the link to our manuscript and fluid intake under high sucrose conditions. 

[Reviewer comment] 3) A potential issue is how FGF21 levels were measured. The authors need to provide more information about this electrochemiluminescence assay. What part of FGF21 does this assay measure. As the authors may know, FGF21 is cleaved and inactivated on the C-terminus. Thus, did the authors measure total or “active” FGF21? These issues should also be addressed in the discussion as potential limitation of the results.

 [Author response] Thank you for raising this relevant issue of which form (total or active) of FGF21 was targeted by the assay used in this study. The FGF21 assay performed here was a commercially available electrochemiluminescence assay produced and supplied by Mesoscale Diagnostics (MSD). We contacted MSD in order to answer your query. MSD informed us that it has, unfortunately, not been tested in-house by MSD as to which form of FGF21 is targeted by this human assay. However, the calibrator used in this assay is full length human FGF21 expressed in E.coli with a predicted molecular mass of 20 kDa. Thus it is likely, though not guaranteed, that we measured total FGF21. We have added in the manuscript the following to account for the ambiguity:

“Additionally, the kit we used to analyse FGF21 does not specify whether total or active FGF21 is measured; considering its calibration using full-length FGF21 with a predicted molecular mass of 20 kDa, it is most likely we measured active FGF21, which has been shown to the primary circulating form of FGF21 [27].” (lines 272-275)

[Author response] We have done this and believe our figure meets PLoS ONE specifications.

---

## [Decision Letter · Decision Letter 1]

10 Jun 2020

PONE-D-20-06810R1

The effect of hydration status on plasma FGF21 concentrations in humans

PLOS ONE

Dear Dr. Carroll,

Thank you for submitting your manuscript to PLOS ONE. After careful consideration, we feel that it has merit but does not fully meet PLOS ONE’s publication criteria as it currently stands. Therefore, we invite you to submit a revised version of the manuscript that addresses the points raised during the review process.

We look forward to receiving your revised manuscript.

Kind regards,

Jo Edward Lewis, Ph.D

Academic Editor

PLOS ONE

Reviewers' comments:

Reviewer's Responses to Questions

**Comments to the Author**

1. If the authors have adequately addressed your comments raised in a previous round of review and you feel that this manuscript is now acceptable for publication, you may indicate that here to bypass the “Comments to the Author” section, enter your conflict of interest statement in the “Confidential to Editor” section, and submit your "Accept" recommendation.

Reviewer #3: (No Response)

2. Is the manuscript technically sound, and do the data support the conclusions?

Reviewer #3: Yes

3. Has the statistical analysis been performed appropriately and rigorously? 

Reviewer #3: Yes

4. Have the authors made all data underlying the findings in their manuscript fully available?

Reviewer #3: Yes

5. Is the manuscript presented in an intelligible fashion and written in standard English?

Reviewer #3: Yes

6. Review Comments to the Author

Reviewer #3: Interesting sub analyis, although with some limitations

The present is a crossover RCT: this should be added in the title,. I would write; a sub analysis of a crossover RCT

Methods: tables should be put at end and not in the text

Methods: sample size calculation was not performed for this rct: this should be put in limitation. Moreover this sub analysis was not prespecificed and also this should be added in limitation section

7. PLOS authors have the option to publish the peer review history of their article (what does this mean?). If published, this will include your full peer review and any attached files.

Reviewer #3: Yes: Fabrizio D'Ascenzo

---

## [Author Response · Author response to Decision Letter 1]

13 Jun 2020

[Reviewer comment] Interesting sub analyis, although with some limitations. The present is a crossover RCT: this should be added in the title,. I would write; a sub analysis of a crossover RCT

[Author response] Thank you for this helpful comment. We have changed the title to “The effect of hydration status on plasma FGF21 concentrations in humans: A subanalysis of a randomised crossover trial”. 

[Reviewer comment] Methods: tables should be put at end and not in the text

[Author response] We followed PLoS author guidelines and inserted tables immediately after the paragraph they were first cited.

[Reviewer comment] Methods: sample size calculation was not performed for this rct: this should be put in limitation. 

[Author response] Lines 264-270 discussed that the study was powered for glycaemia not FGF21. We have added the following to this paragraph to emphasise the point: 

“It may be that there is a small but meaningful response, which we may not have been able to detect due to lack of statistical power for FGF21 as the outcome” (line 266)

[Reviewer comment] Moreover this sub analysis was not prespecificed and also this should be added in limitation section

[Author response] We hope it was clear this was not a pre-specified outcome in our introduction, but have reiterated this point in our discussion:

“Nonetheless, future work should investigate the postprandial FGF21 response according to hydration status; this is particularly important considering our present analysis was decided post hoc and therefore not pre-specified in our research plan or trial registration.” (lines 256-258)

Additional edit: Since resubmitting the manuscript, the first author has changed institutional affiliation, and this has been amended on the title page.

---

## [Editor Report · Decision Letter 2]

18 Jun 2020

The effect of hydration status on plasma FGF21 concentrations in humans: A subanalysis of a randomised crossover trial

PONE-D-20-06810R2

Dear Dr. Carroll,

We’re pleased to inform you that your manuscript has been judged scientifically suitable for publication and will be formally accepted for publication once it meets all outstanding technical requirements.

Kind regards,

Jo Edward Lewis, Ph.D

Academic Editor

PLOS ONE
---

## [Editor Report · Acceptance letter]

23 Jul 2020

PONE-D-20-06810R2 

The effect of hydration status on plasma FGF21 concentrations in humans: A subanalysis of a randomised crossover trial 

Dear Dr. Carroll:

I'm pleased to inform you that your manuscript has been deemed suitable for publication in PLOS ONE. Congratulations! Your manuscript is now with our production department. 

Kind regards, 

on behalf of

Dr. Jo Edward Lewis 

Academic Editor

PLOS ONE